# Protective Effect of Ectoin on UVA/H$_2$O$_2$-Induced Oxidative Damage in Human Skin Fibroblast Cells

Wenjing Cheng [1,2,†], Quan An [3,†], Jiachan Zhang [1,2,*], Xiuqin Shi [1,2], Changtao Wang [1,2,*], Meng Li [1,2] and Dan Zhao [1,2]

1   Beijing Key Laboratory of Plant Resource Research and Development, College of Chemistry and Materials Engineering, Beijing Technology and Business University, Beijing 100048, China
2   Institute of Cosmetic Regulatory Science, Beijing Technology and Business University, Beijing 100048, China
3   Yunnan Baiyao Group Co., Ltd., Kunming 650000, China
*   Correspondence: xiaochan8787@163.com (J.Z.); wangct@btbu.edu.cn (C.W.); Tel.: +86-13426258535 (J.Z.); +86-18910839671 (C.W.)
†   These authors contributed equally to this work.

**Abstract:** Ectoin is an amino acid derivative that can create a balance between the osmotic pressure of cells and can protect enzymes, DNA proteins, and nucleic acids under extreme conditions. Ectoin has also been reported to slow skin aging. However, there are few reports on the protective effect of Ectoin on oxidative damage, especially on the regulation of PI3K/AKT-pathway-related genes at the mRNA level. UVA-induced oxidative stress injury and H$_2$O$_2$-induced oxidative stress injury are two common oxidative stress injury models. Skin fibroblasts produce a large number of ROS following excessive UV radiation or oxidative stimulation by H$_2$O$_2$, which further inhibits cell proliferation and causes cell apoptosis. In this study, UVA- and H$_2$O$_2$-induced oxidation models of human skin fibroblasts were established separately to investigate the protective effect of Ectoin. Further studies on the mechanisms involved, for example, the expression levels of genes associated with the PI3K/AKT signaling pathway and levels of antioxidant enzymes in cells, were determined. We found that Ectoin upregulated genes associated with the PI3K/AKT signaling pathway, including COL1A1, COL1A2, FN1, IGF2, NR4A1, and PIK3R1, but decreased intracellular ROS levels and malondialdehyde (MDA), while increasing the activities of superoxide dismutase (SOD) and glutathione peroxidase (GSH-Px). In conclusion, our results indicate that Ectoin exerts protective properties by the upregulated genes COL1A1, COL1A2, FN1, IGF2, NR4A1, and PIK3R1 and upregulating antioxidative enzyme levels.

**Keywords:** Ectoin; oxidative damage; H$_2$O$_2$; UVA; PI3K/AKT signaling pathway

## 1. Introduction

Fibroblasts play a major role in supporting many tissue structures and are important cells in the dermis that can secrete cytokines and thus participate in multiple healing stages [1–3]. Human skin fibroblasts (HSFs) are the main cell component in the dermis of human skin and form the main body of the dermis together with its own secreted elastic fibers, collagen fibers, and matrix components [4]. Numerous studies have demonstrated that HSF cells are important for the skin aging process [5–7]. UVA-induced oxidative stress damage and H$_2$O$_2$-induced oxidative stress damage are two common models of oxidative stress damage. When UVA/H$_2$O$_2$ stimulates HSF cells, the cells undergo oxidative injury, and the composition of antioxidant-related enzymes in the cells changes. Therefore, scientific research often establishes oxidative damage models to evaluate the efficacy of active substances. Fibroblasts produce a large amount of active oxygen following excessive ultraviolet radiation based on the dose of ultraviolet light, while H$_2$O$_2$ is the main component of active oxygen [8,9]. At very high concentrations, reactive oxygen species inhibit cell proliferation and lead to cell apoptosis [10]. UVA and H$_2$O$_2$ eventually cause oxidative stress damage to the skin [11–14].

Protecting the tissues from unwanted side effects is one strategy to lessen the unfavorable consequences of UVA and $H_2O_2$ in medical practice. As a result, the therapeutic benefits of natural compounds with antioxidant capabilities that may lessen the severity of UVA and $H_2O_2$-induced toxicities could be advantageous. Antioxidants are widely utilized as nutrients, and their effectiveness in reducing tissue and organ toxicity from various medicines was studied. Numerous studies have looked at the effectiveness of antioxidants as nutrients in reducing tissue and organ toxicity caused by a variety of conditions. Reactive oxygen species damage can be avoided and repaired by endogenous and exogenous antioxidants (ROS). They are referred to as "free radical scavengers" because they can strengthen the immune system and reduce the danger of sickness and toxins. Superoxide and other peroxides are chelated by enzyme-based antioxidants, such as catalase (CAT), glutathione peroxidase (GPx), and superoxide dismutase (SOD). They serve as defense mechanisms against endogenous antioxidants, clearing ROS activity and buildup in cells and preserving redox balance [15–18].

The PI3K/AKT signaling pathway is implicated in many key cell functions, such as protein synthesis, cell cycle, apoptosis, drug-resistant growth factors (EGF, NGF, and VEGF), hormones (prostaglandins and PGE2), and cytokine stimulation [19–21]. Collagen type I $\alpha$ 1 chain (COL1A1) and Collagen type I $\alpha$ 2 chain (COL1A2) encode the front $\alpha$1 and $\alpha$2 chains of type I collagen [22]. Collagen type IV $\alpha$ 5 chain (COL4A5) encodes one of the six subunits of type IV collagen, which is the main structural component of the basement membrane. Fibronectin 1 (FN1) is a kind of high-molecular-weight glycoprotein involved in a variety of biological functions, and its overexpression inhibits cell apoptosis by activating the PI3K/AKT signaling pathway [23,24].

Ectoin is an amino acid derivative that can balance the osmotic pressure of cells and can also protect enzymes, DNA proteins, and nucleic acids under extreme conditions, such as high levels of radiation [25]. Ectoin can reduce the effects of UVA-induced damage in human keratinocytes and fibroblasts by preventing the release of secondary messengers, the activation of AP-2, the expression of intercellular adhesion molecule-1, and the mutation of mitochondrial DNA [26]. Researchers suggest that Ectoin may also play an important role in increasing the activity of antioxidant enzymes and the level of nonenzymatic antioxidants. Research also shows that Ectoin plays an important role in stabilizing the cell membrane, reducing inflammation, and mitigating DNA damage induced by ultraviolet, infrared, and visible radiations. However, the detailed mechanisms involved in these processes have still not been fully understood [27,28].

Topically applied to the skin, Ectoin not only exerts a cellular protective effect but also exerts a skin-protective barrier function. Heinrich et al. assessed an Ectoin formulation with regard to its antiaging properties and found that 2% Ectoin was more effective in terms of skin hydration, skin elasticity, and skin surface structure than the vehicle treatment [29]. Furthermore, Ectoin can be used for the treatment of mild to moderate atopic dermatitis by improving skin moisture and strengthening skin barrier function [30,31].

However, the skin-protective effects of Ectoin are well documented, and the underlying mechanisms have still not been fully understood. There is a lack of reports that discuss insight into the mechanisms of Ectoin, especially changes in the expressions of genes involved in the PI3K/AKT pathway.

In this study, we established UVA-induced and $H_2O_2$-induced oxidative models of human skin fibroblast cells separately, discuss the protective effects exerted by Ectoin in the two models, and further studied the molecular mechanisms. The cellular levels of the antioxidant enzymes and oxidative products were also measured. Overall, this study provides a theoretical basis for the application of Ectoin for the treatment of antiskin oxidative damage.

## 2. Materials and Methods

### 2.1. Materials

HSFs were bought from the Cell Resource Center, Institute of Basic Medicine, Chinese Academy of Medical Sciences. Ectoin (purity $\geq$ 98%; Bloomage Biotechnology Co., Ltd., Beijing, China), Dulbecco's Modified Eagle Medium (DMEM), fibroblast medium, fetal bovine serum (FBS), phosphate buffer saline (PBS), penicillin, streptomycin, and 0.25% (with EDTA) trypsin were all bought from Gibco Life Technologies (Carlsbad, CA, USA). RIPA buffer total protein extract (Sinogene, Beijing, China), $100\times$ phosphatase inhibitors (Sinogene, Beijing, China), Bradford protein assay reagent (Sinogene, Beijing, China), and a TransStart® Top Green qPCR SuperMix kit were obtained from Beijing TransGen Biotech Co., Ltd. ROS, MDA, SOD, and GSH-Px kits were bought from Beyotime Biotechnology Co., Ltd.

All other chemicals were of analytical grade or complied with the standards required for cell culture experiments.

### 2.2. Cell Culture

Cells were cultured in DMEM, supplemented with 10% FBS, 1% fibroblast growth additives, and 1% penicillin ($1 \times 10^5$ U/L)–streptomycin (100 mg/L). The cells were incubated at 37 °C in a humidified atmosphere with 5% $CO_2$. The culture medium was renewed when cell confluence reached 80%, and the cells were digested using 0.25% (with EDTA) trypsin.

### 2.3. Cell Viability

Cell viability was determined using the colorimetric CCK-8 method [32–34], following the manufacturer's instructions. HSF cells were seeded into a 96-well plate at a concentration of $1 \times 10^5$ cells per well at 37 °C and were incubated in a cell incubator (Shanghai Shengke Instrument Equipment Co., Ltd., Shanghai, China) with 5% $CO_2$ for 12 h. The cells were treated with Ectoin at different concentrations ranging from 8 to 500 µg/mL for 24 h.

### 2.4. UVA and $H_2O_2$-Induced Model Establishment

UVA-induced model establishment: HSF cells were seeded into each well of a 96-well plate at a density of $1 \times 10^4$ cells and cultured in a cell incubator for 12 h. The medium was replaced with 100 µL of PBS (pH 7.4, 0.01 M), and the cells were stimulated under different doses of UVA ranging from 7 to 25 J/cm$^2$. UVA (365 nm) radiation was created using an UVA lamp (T8 40W, Royal Dutch Philips Electronics Ltd., Eindhoven, The Netherlands), and intensity was measured using an UV power meter (LS125, Shenzhen Shanglin Technology Co., Ltd., Shenzhen, China). $IC_{50}$ parameters were chosen to establish the UVA-induced model.

$H_2O_2$-induced model establishment: The $H_2O_2$-induced model was established using the same method described above using different concentrations of $H_2O_2$ (250–3000 µmol·L$^{-1}$) instead of PBS for different periods ranging up to 0.5 h. $IC_{50}$ parameters were chosen to establish the $H_2O_2$-induced model.

The HSF cell viability was determined using the CCK-8 (Biorigin (Beijing) Inc., Beijing, China) method. Images of the cells were captured under a microscope (Shanghai Tucsen Vision Technology Co., Shanghai, China) to observe morphological changes.

### 2.5. Protective Effects of Ectoin on UVA and $H_2O_2$-Induced Model

The cells were inoculated into 96-well plates at a density of $1 \times 10^4$ and kept for 12 h to establish the UVA-induced or $H_2O_2$-induced oxidative models.

Then, different concentrations of Ectoin ranging from 8 to 500 µg/mL were added and incubated for different durations (24 h). Cell viability was calculated using the CCK-8 method.

*2.6. RT-qPCR*

The cells were collected after being subjected to the different treatments. Total RNA was extracted using the TriQuick Reagent (Beijing Solarbio Science and Technology Co., Ltd., Beijing, China), and the cDNA was synthesized following the manufacturer's instructions. Reverse transcription-qPCR (RT-qPCR) was performed according to the instructions given in the TransStart® Top Green qPCR SuperMix kit. The reaction system was 20 µL in total. The specific reagents and dosages used are presented in Table S1, while the primers used are presented in Table 1. GAPDH was used as a normalization control.

**Table 1.** Primer sequences for real-time PCR.

| Primer Name | Direction | Primer Sequences (5′-3′) |
|:---:|:---:|:---:|
| GAPDH | F | TCAGACACCATGGGGAAGGT |
|  | R | TCCCGTTCTCAGCCATGTAG |
| COL4A5 | F | CAAGGTCTACCAGGTCCAGAA |
|  | R | TCATTCCATTGAGACCCGGC |
| FN1 | F | CCCAATTGAGTGCTTCATGCC |
|  | R | CCTCCAGAGCAAAGGGCTTA |
| IGF2 | F | TCCTGTGAAAGAGACTTCCAG |
|  | R | GTCTCACTGGGGCGGTAAG |
| COL1A1 | F | GAGGGCCAAGACGAAGACATC |
|  | R | CAGATCACGTCATCGCACAAC |
| COL1A2 | F | GTTGCTGCTTGCAGTAACCTT |
|  | R | AGGGCCAAGTCCAACTCCTT |
| PIK3R1 | F | ACCACTACCGGAATGAATCTCT |
|  | R | GGGATGTGCGGGTATATTCTTC |
| NR4A5 | F | ATGCCCTGTATCCAAGCCC |
|  | R | GTGTAGCCGTCCATGAAGGT |

The cyclic parameters used were predenaturation at 94 °C for 30 s and PCR reaction (45 cycles of 94 °C for 15 s, 60 °C for 15 s, and 72 °C for 10 s), and fluorescence data were collected at 72 °C. Analyses of the relative gene expression levels were performed using the $2^{-\Delta\Delta CT}$ method [35].

*2.7. Determination of Intracellular Reactive Oxygen Species (ROS)*

The production of intracellular ROS was determined using a fluorescence microscopy kit (Shanghai Tulsen Vision Technology Co., Ltd., Shanghai, China), following the manufacturer's instructions. Then, 2′,7′-dichlorofluorescein diacetate (DCFH2-DA), which is readily taken up by cells and can be de-esterified to 2′,7′-dichlorodihydrofluorescein (DCFH2), was subsequently used. Thereafter, DCFH2 can be oxidized by intracellular ROS to form fluorescent 2′,7′-dichlorofluorescein (DCF). The cells were treated with different concentrations of Ectoin, which affected ROS levels and finally induced fluorescent changes.

*2.8. Determination of Intracellular GSH-Px, MDA, and SOD*

The cells subjected to the different treatments were collected. GSH-Px, MDA, and SOD levels were detected using colorimetric reagent kits (Nanjing Jiancheng, Nanjing, China) following the manufacturer's instructions. Ascorbic acid (60 µg/mL; Supplementary Figures S1 and S2) was chosen as a positive control. The cell viability after ascorbic acid treatment is given in the supplementary materials.

*2.9. Statistics*

All experiments were performed in triplicate at least, and data were expressed as mean ± standard deviation (SD). The data were analyzed using SPSS 17.0 (SPSS, Armonk, NY, USA) and GraphPad Prism 9.0 (GraphPad Software, San Diego, CA, USA) software.

One-way analysis of variance was used to determine the significance of differences between the groups (#, * $p < 0.05$; ##, ** $p < 0.01$).

## 3. Results

### 3.1. Cell Viability of Ectoin and Oxidative Stress Damaged Cells

UVA/$H_2O_2$ can decrease cell viability and cause the oxidative damage of cells. As the dose of UVA or $H_2O_2$ was increased, the survival rate of the HSF cells gradually decreased (Figure 1).

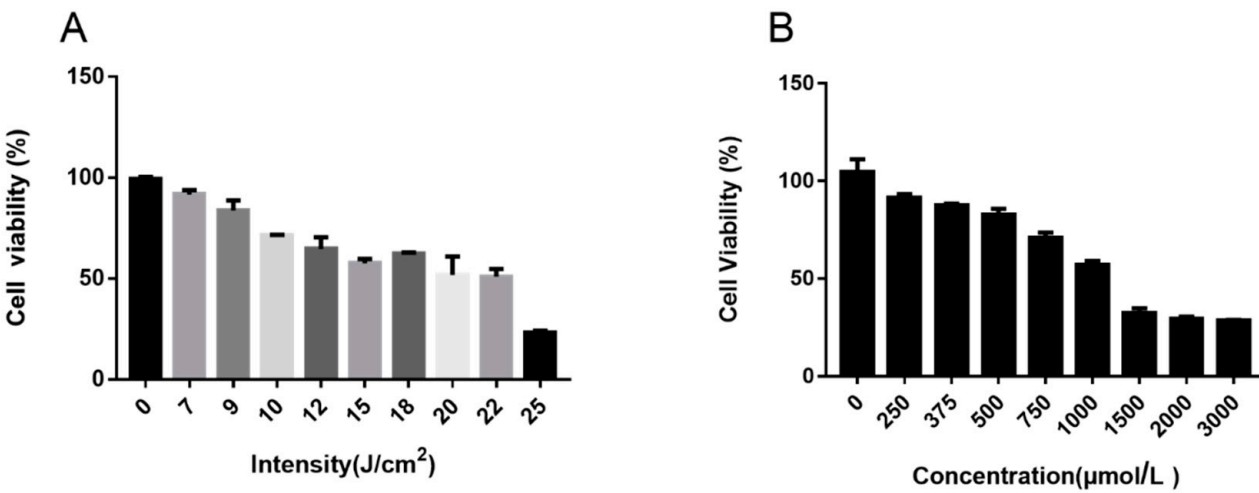

**Figure 1.** The toxicity of UVA (**A**) and $H_2O_2$ (**B**) to HSF cells. The results are expressed as mean $\pm$ SD (n = 3). Different superscripts mean significant difference (uppercase for 24 h). There are six replicates in each group.

Figure 1A shows the survival rates of the HSF cells after treatment with different doses of UVA. When the UVA radiation dose was 22 J/cm$^2$, the cell survival rate was $(50.89 \pm 3.84)$%. Figure 1B shows the survival rate of the HSF cells treated with 250–3000 μmol·L$^{-1}$ $H_2O_2$ for 0.5 h. After treatment with 1000 μmol·L$^{-1}$ $H_2O_2$ for 0.5 h, the cell survival rate decreased to $(57 \pm 2.03)$%. Therefore, we chose to use IC$_{50}$ parameters to establish the damage models. UVA at a dose of 20 J/cm$^2$ was used to establish the UVA-induced oxidative stress damage model. The modeling condition of the $H_2O_2$-induced oxidative stress damage model was 1000 μmol·L$^{-1}$ $H_2O_2$ for 0.5 h.

Ectoin protects cells from damage by UVA/$H_2O_2$ and improves cell viability. Ectoin at a concentration ranging from 8 to 500 μg/mL is nontoxic to HSF cells (Figure 2A) and is able to exert a good promotion effect on cell proliferation. Ectoin at a concentration ranging from 8 to 125 μg/mL for 24 h was able to significantly promote the proliferation of the HSF cells in a time-dependent manner. Two different concentrations (8 and 16 μg/mL for 24 h, separately) were used in subsequent experiments. Both 8 and 16 μg/mL of Ectoin enhanced cell viability under UVA or $H_2O_2$ stimulation, and there was no significant difference in cell viability between either of the two concentrations of Ectoin (Figure 2B,C).

Therefore, to facilitate the comparison of the protective effect of the same concentration of Ectoin exerted during UVA/$H_2O_2$ oxidative damage, 8 μg/mL of Ectoin was selected as the treatment concentration to be used on subsequent UVA and $H_2O_2$ damage models.

### 3.2. Antioxidant Enzymes and ROS Levels of Oxidative Stress Damaged Cells

As shown in Figure 3A,B,D,E, the SOD and GSH-Px vitalities reduced significantly following UVA/$H_2O_2$ damage (both $p < 0.05$), while MDA levels increased significantly, compared with the UVA-induced or $H_2O_2$-induced models (both $p < 0.05$, Figure 3C,F). Both Ectoin and ascorbic acid could significantly increase the SOD and GSH-Px activities, which had decreased due to UVA/$H_2O_2$ damage ($p < 0.05$, Figure 3A,B,D,E). In addition, there was no significance difference between the abilities of Ectoin and ascorbic acid to

increase SOD activity, compared with the UVA-induced model. Moreover, Ectoin exerted a stronger increasing effect ($p < 0.01$, Figure 3B) on the UVA-induced decrease in GSH-Px levels than ascorbic acid.

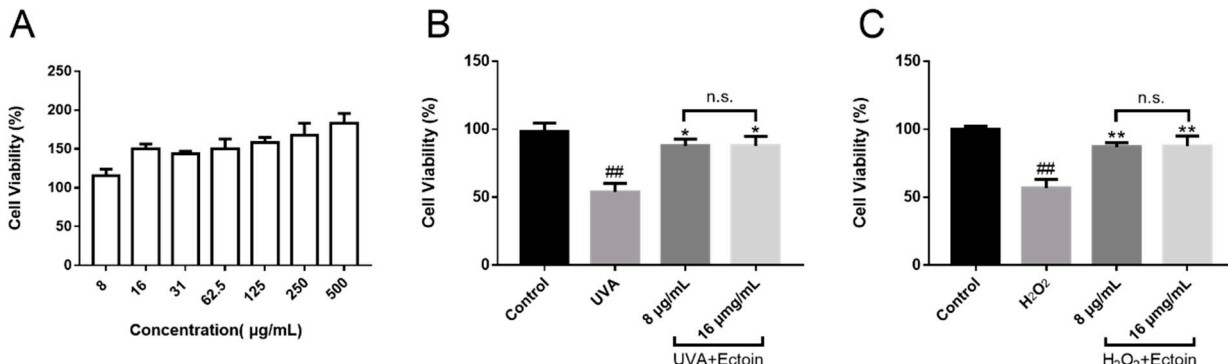

**Figure 2.** The toxicity of Ectoin (**A**) to HSF cells and the repair effects of Ectoin on UVA-induced (**B**) and $H_2O_2$-induced (**C**) oxidative stress damage in HSF cells. The results are expressed as mean ± SD (n = 3). The model in (**B**) was established by UVA ($22$ J/cm$^2$), and the model in (**C**) was established by $H_2O_2$ ($1000$ μmol·L$^{-1}$). The discussed concentrations of Ectoin are 8 and 16 μg/mL. Statistical significance was determined by ANOVA test. (## $p < 0.01$, compared with the DMEM-treated control. * $p < 0.05$, and ** $p < 0.01$, compared with the Model. n.s., $p > 0.05$).

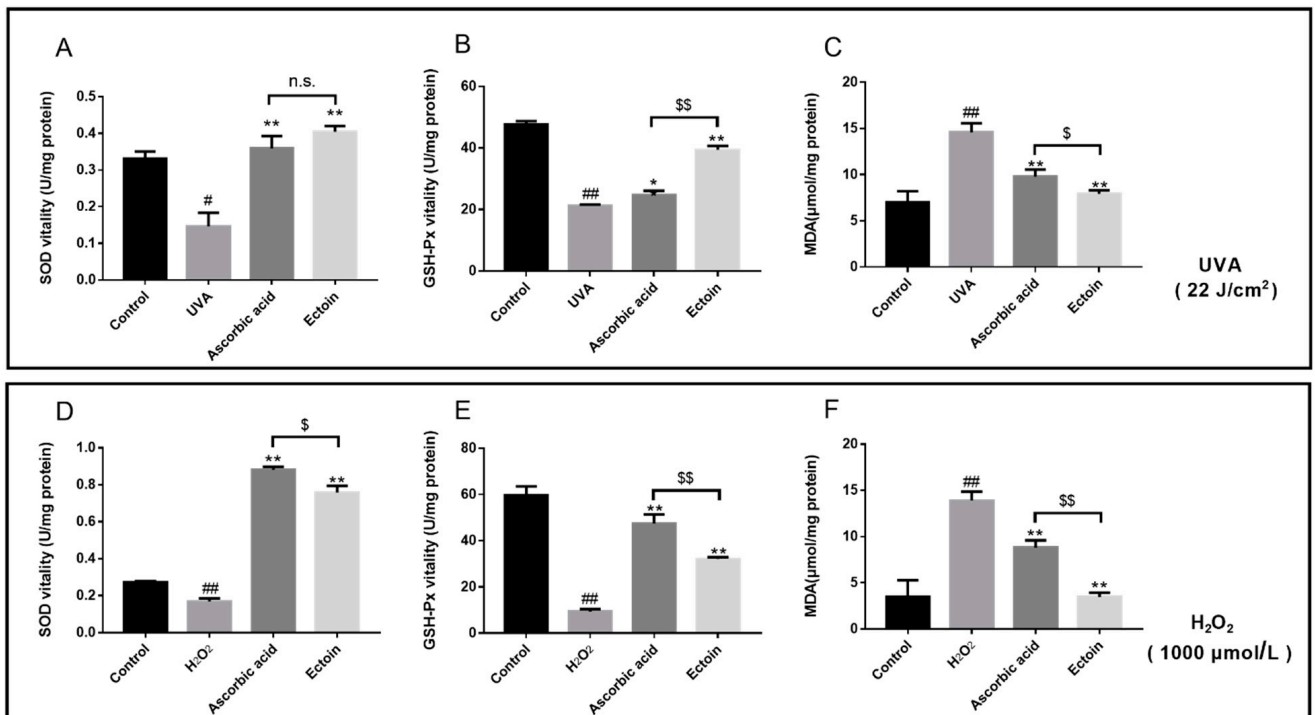

**Figure 3.** The antioxidant activities of Ectoin on the total SOD activities (**A**,**D**), GSH-Px activities (**B**,**E**), MDA contents (**C**,**F**). The uppercase (**A**–**C**) represents the UVA-induced damaged study, while the lowercase (**D**–**F**) represents the $H_2O_2$-induced damaged study. Ascorbic acid (60 μg/mL, 24 h) was chosen as the positive control. The model in (**A**–**C**) was established by UVA ($22$ J/cm$^2$), and the model in (**D**–**F**) was established by $H_2O_2$ ($1000$ μmol·L$^{-1}$). Statistical significance was determined by ANOVA test. (## $p < 0.01$, and # $p < 0.05$, compared with the Control. * $p < 0.05$, and ** $p < 0.01$, compared with the Model. \$ $p < 0.05$, and \$\$ $p < 0.01$, compared with the Ascorbic acid).

In addition, Ectoin and ascorbic acid caused a significant change in reducing MDA levels, compared with the UVA-induced model. A more significant decrease in Ectoin

treatment was observed on the UVA/$H_2O_2$-induced accumulation of MDA, compared with the positive control (Figure 3F).

UVA/$H_2O_2$ treatment could induce the production of cellular ROS. The fluorescent changes represented showed the changes in ROS levels. Fluorescence intensity was observed to have increased along with the formation of ROS. After UVA/$H_2O_2$ stimulation, the release of ROS in the cells of the model groups increased (Figure 4A,C). DCF fluorescence intensity is shown in Figure 4B,D. Then, a decrease in cell fluorescence intensity was observed after the damaged cells were treated with Ectoin or ascorbic acid, which indicates that Ectoin could reduce ROS levels.

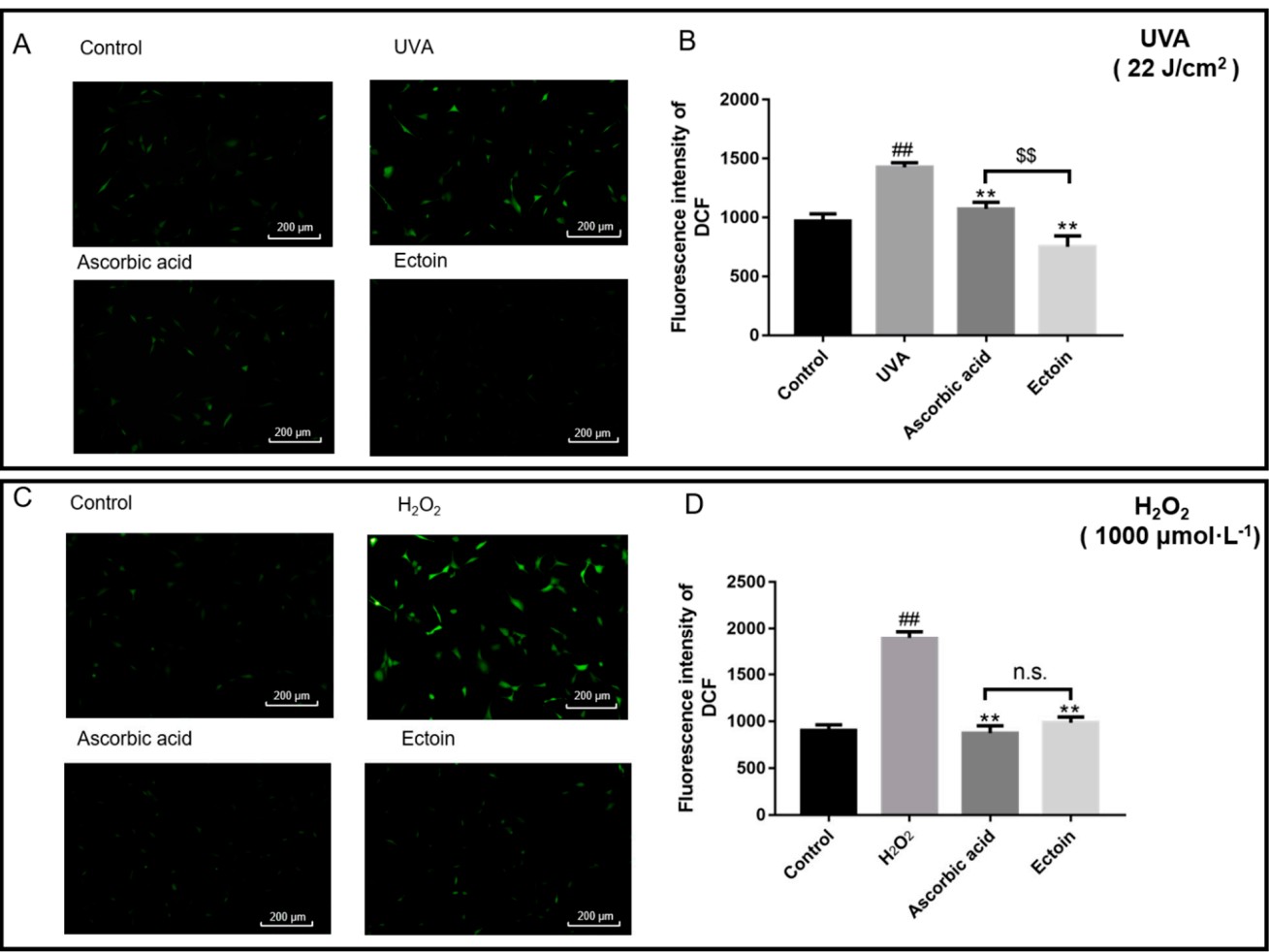

**Figure 4.** (**A**,**B**) Effects of Ectoin and ascorbic acid on ROS production induced by UVA/$H_2O_2$ stimulation; (**C**,**D**) effects of Ectoin and ascorbic acid on ROS content in HSF cells stimulated by UVA/$H_2O_2$. Statistical significance was determined by ANOVA test. (## $p < 0.01$, compared with the Control. ** $p < 0.01$, compared with the Model. n.s., $p > 0.05$).

### 3.3. The Effects of Ectoin on the Expression of Related Genes in the PI3K/AKT Signaling Pathway

The UVA/$H_2O_2$-injured cells were treated with Ectoin for 24 h, and then the expression levels of COL1A1, COL1A2, COL4A5, FN1, IGF2, NR4A1, and PIK3R1 were detected. The results are shown in Figure 5. It can be observed that all seven genes were significantly downregulated after UVA injury or $H_2O_2$ injury (Figure 4).

Ectoin increased the expression levels of COL1A1 (Figure 5A), COL1A2 (Figure 5B), FN1 (Figure 5D), IGF2 (Figure 5E), NR4A1 (Figure 5F), and PIK3R1 (Figure 5G) in the model induced by UVA irradiation. However, the expression of COL4A5 (Figure 5C) decreased after Ectoin treatment.

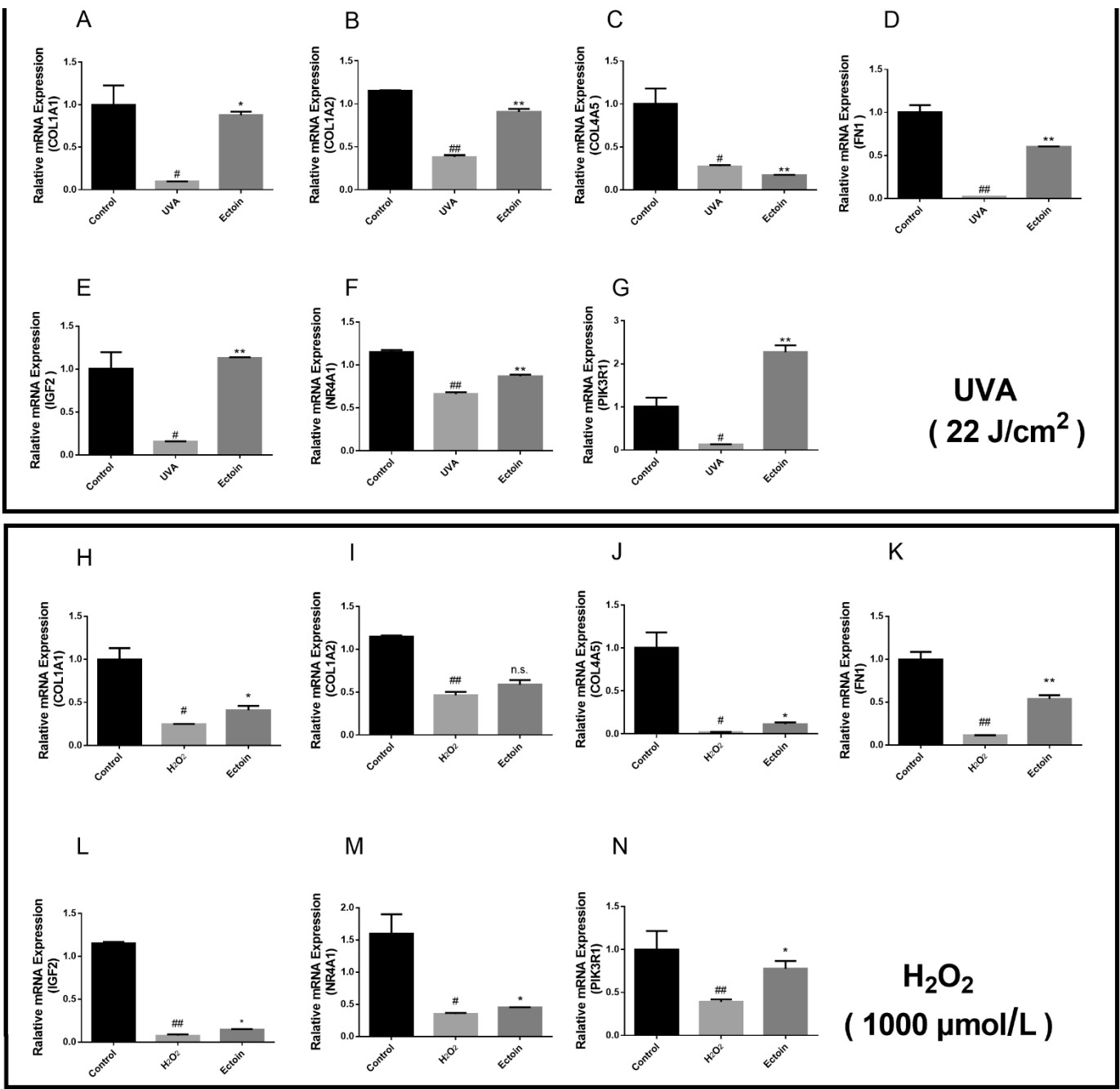

**Figure 5.** The relative gene expression changes after the repair of Ectoin on UVA and $H_2O_2$ injury. Genes, COL1A1 (**A,H**), COL1A2 (**B,I**), COL4A5 (**C,J**), FN1 (**D,K**), IGF2 (**E,L**), NR4A1 (**F,M**), and PIK3R1 (**G,N**) were measured. The model in (**A–G**) was established by UVA (22 J/cm$^2$), and the model in (**H–N**) was established by $H_2O_2$ (1000 µmol·L$^{-1}$). Statistical significance was determined by ANOVA test. (## $p < 0.01$, and # $p < 0.05$, compared with the Control. * $p < 0.05$, and ** $p < 0.01$, compared with the Model. n.s., $p > 0.05$).

In the model induced by $H_2O_2$, Ectoin increased the expression levels of COL1A1 (Figure 5H), COL4A5 (Figure 5J), FN1 (Figure 5K), IGF2 (Figure 5L), NR4A1 (Figure 5M), and PIK3R1 (Figure 5N). There was no significance of the time when Ectoin was added to the injured cells induced by $H_2O_2$. The PI3K/AKT pathway was accelerated and caused upregulation of the downstream genes, which played a role in the response to oxidative damage [36,37].

## 4. Discussion

One of the main causes of skin aging is oxidative damage to the skin [38]. ROS are oxygen-containing small species, such as $^1O_2$, $O_3$, $OH\bullet$, $H_2O_2$, and $O_2\bullet-$ [39]. Previous studies have shown that a large amount of ROS production can lead to oxidative stress, and oxidative damage can be caused when excessive ROS cannot be properly removed [39]. $H_2O_2$ is the main ROS, and it can directly or indirectly cause cell injury and induce cell death [40,41]. In addition, ultraviolet radiation is another important source of ROS production. Long-term exposure to ultraviolet radiation will damage the structure and function of DNA and proteins and cause other macromolecular damages. Although UVB energy is greater, UVA light can penetrate deeper into the skin to cause DNA damage. Much effort has been made to fight against skin aging as focus on cosmetic beauty has increased [42]. Therefore, $H_2O_2$ and UVA can cause oxidative damage by stimulating cells to produce excessive ROS.

Ectoin is a natural vital substance that was developed for cosmetic applications [29]. A study conducted using an UVA stress model showed that Ectoin protects the skin from the harmful effects of UVA-induced cell damage in a number of different ways. The study showed that UVA-induced secondary messenger release, transcription factor AP-2 activation, intercellular adhesion molecule-1 expression, and mitochondrial DNA mutation could be prevented by Ectoin [26]. We hypothesized that Ectoin could reduce the ROS produced by $H_2O_2$ stimulation, thus achieving the protective effect of cells. Furthermore, we evaluated the protective effects of Ectoin using both UVA-induced and $H_2O_2$-induced oxidative HSF models. Ectoin significantly increased the cell viability of the damaged cells, indicating the effect of Ectoin in promoting HSF cell proliferation. The PI3K/AKT pathway plays an important protective role against oxidative stress and resulting apoptosis [43]. PI3Ks are members of the lipid kinase family [36] and are activated by growth factor receptors, cell adhesion molecules, and G-protein-coupled receptors [44]. The PI3K/AKT signaling pathway is involved in many biological processes, including cell proliferation, apoptosis, angiogenesis, and glucose metabolism [45–47]. In our study, the expression levels of COL1A1, COL1A2, COL4A5, FN1, IGF2, NR4A1, and PIK3R1 were detected using RT-qPCR. Genes, such as IGF2, NR4A1, and PIK3R1, were upregulated, accelerating the PI3K/AKT pathway and downstream genes, such as COL1A1, COL1A2, COL4A5 (not in the UVA model), and FN1. The results showed that Ectoin could regulate the PI3K/AKT pathway at the molecular level. Resveratrol, an antioxidant additive, can partially inhibit oxidative stress and induce apoptosis by activating the PI3K/AKT pathway [48]. Zhang et al. found that Angelica polysaccharide can promote the activation of the PI3K/AKT signaling pathway, improve cell viability, reduce cell apoptosis and ROS production, and thus reduce the level of cell oxidative damage caused by $H_2O_2$ at a concentration of 300 μM [49]. Furthermore, the antioxidant enzymes play an important role in the defense against oxidative damage. Apart from the enzymes (such as SOD, CAT, and GSH-Px), ROS and MDA level are usually discussed in related studies. In this study, ROS, MDA, SOD, and GSH-Px were selected to represent levels of oxidative stress in cells. The MDA content of the experimental group treated with Ectoin decreased significantly, while levels of SOD and GSH-Px showed a significant increase, proving that Ectoin protects cells from UVA and $H_2O_2$ oxidative damage. Many potential functional materials play a role in accelerating the activities of antioxidant enzymes and decreasing levels of peroxide products. Skin-derived precursors (SKPs) significantly reduced the levels of the UV-induced apoptosis of cutaneous cells in the 3D skin model, which decreased the ROS and MDA levels and increased the GPX, SOD, and CAT levels [37]. In addition, the curcumin-treated groups showed a decrease in ROS and MDA content [50].

In our research study, Ectoin activated the genes in the PI3K/AKT pathway and enhanced the levels of antioxidant enzymes to protect HSF cells from oxidative stress, the damage it mediates, and even apoptosis. Our research provides a direction for the development of antioxidants and other active substances that protect against oxidative damage.

## 5. Conclusions

The results indicate that Ectoin can protect cells from oxidative damage caused by UVA and $H_2O_2$ by regulating antioxidant-related enzymes and upregulating genes involved in the PI3K/AKT signaling pathway and promoting antioxidant stress resistance of skin fibroblasts. Overall, this study provides a theoretical basis for the application of Ectoin as an antioxidant used in cosmetics. In general, the results of this study can be used as a basis to reduce damage caused by oxidative stress in the skin. However, more research is needed to identify other potential mechanisms of Ectoin action.

**Supplementary Materials:** The following supporting information can be downloaded at: https://www.mdpi.com/article/10.3390/app12178531/s1, Figure S1: Effects of different concentrations of Ascorbic acid on cell viability (n = 6).; Figure S2: The protective effects of Ascorbic acid on UVA-induced (A) and $H_2O_2$-induced (B) oxidative stress damage in HSF cells. Table S1: Reagents and dosage.

**Author Contributions:** Conceptualization, W.C. and Q.A.; methodology, M.L.; software, X.S.; validation, W.C., Q.A. and D.Z.; formal analysis, C.W.; investigation, M.L.; resources, Q.A.; data curation and visualization, W.C. and Q.A.; writing of the manuscript, W.C.; writing—review and editing, W.C. and Q.A.; supervision, J.Z.; project administration, J.Z. and C.W. All authors have read and agreed to the published version of the manuscript.

**Funding:** Zhejiang Province Leading Innovation and Entrepreneurship Team Project (2020R01018) Zhejiang Provincial Science and Technology Department, Shaoxing City "Top Ranking" System Science and Technology Project (2021B42001) Zhuji Science and Technology Bureau, and Zhejiang Province Science and Technology Plan Project (2022C02037) Science and Technology Department of Zhejiang Province.

**Institutional Review Board Statement:** Not applicable.

**Informed Consent Statement:** Not applicable.

**Data Availability Statement:** Such information is available from the corresponding author upon reasonable request.

**Acknowledgments:** We would like to thank the members of the panel for their guidance and contribution to this study, and various funds for their support.

**Conflicts of Interest:** The authors have no conflict of interest.

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
