# Peer review of "Protective Effect of Ectoin on UVA/H2O2-Induced Oxidative Damage in Human Skin Fibroblast Cells"

_applsci, doi:10.3390/app12178531_

Round 1

Reviewer 1 Report

(1) Is H2O2 only the measurement of intracellular active oxygen? Or are you also measuring superoxide and hydroxyl radicals? (2) Since ascorbic acid can only remove superoxide, it is doubtful whether it can be used as a control when measuring H2O2 this time. (3) The overall flow is thought to be good.

Author Response

对审稿人意见的详细回复

稿件编号:applsci-1874334

稿件类型:文章

修改标题:Ectoin对UVA/H 2 O 2诱导的人体皮肤成纤维细胞氧化损伤的保护作用

Authors: Wenjing Cheng, Quan An, Jiachan Zhang *, Xiuqin Shi, Changtao Wang *, Meng Li, Dan Zhao

亲爱的审稿人,

First of all, we would like to thank the reviewers for their comments on revising the manuscript, and we are also very grateful to the reviewer for their work on our manuscript. Detailed point-by-point responses to the reviewers’ comments are provided in the following pages. Note that the reviewers’ comments are presented in Italics, and our responses are in Roman and blue font. In addition, we addressed all these major points and other issues carefully and revised the manuscript accordingly, and We highlight the revised parts in the article in red. Please let me know if you have any further questions.

Sincerely,

Jiachan Zhang

Beijing Key Lab of Plant Resource Research and Development, Beijing Technology and Business University, Fucheng Road, Beijing 100048, China

Tel.: +86-13426258535

To the referee’s comments, we make the following responses and changes in the manuscript:

-Is H2O2 only the measurement of intracellular active oxygen? Or are you also measuring superoxide and hydroxyl radicals?

Reply:

Thank you for your comments and great suggestions.

Skin fibroblasts produce a large number of reactive oxygen species (ROS) following excessive UV radiation or oxidative stimulation by H2O2, which further inhibits cell proliferation and causes cell apoptosis [1-7]. It is reported that H2O2 is the main ROS and can directly or indirectly damage cells and induce apoptosis and necrosis [8,9]. As a result, H2O2 is widely used as an inducer for oxidative stress model in vitro, and one of the common indicators to measure the cellular oxidative damage levels is ROS content, which were widely reported in lots of researches [10,11]. While, the measurement of superoxide and hydroxyl radicals in cells were rarely reported, their scavenging measurements in test tube were widely applied. Considering the aim of the study, we decided to measure the cellular ROS contents to evaluate the oxidative damage level.

Thank you again for your suggestions, which give us a new idea in the forward study.

Refs.

  1. Marionnet, C.; Pierrard, C.; Golebiewski, C.; Bernerd, F. Diversity of biological effects induced by longwave UVA rays (UVA1) in reconstructed skin. PLoS One. 2014; 9: e105263.
  2. Zhong, J.; Li, L. Skin-derived precursors against UVB-induced apoptosis via Bcl-2 and Nrf2 up-regulation. Biomed Research International. 2016, 6894743.
  3. Fu, H.; You, S.; Zhao, D.; An, Q.; Zhang, J.; Wang, C.; Wang, D.; Li, M. Tremella fuciformis polysaccharides inhibit UVA-induced photodamage of human dermal fibroblast cells by activating up-regulating Nrf2/Keap1 pathways. Journal of Cosmetic Dermatology. 2021, 20: 4052–4059.
  4. Terra, V.A.; Souza-Neto, F.P.; Pereira, R.C.; Silva, T.N.X.; Costa, A.C.C.; Luiz, R.C.; Cecchini, R.; Cecchini, A.L. Time-dependent reactive species formation and oxidative stress damage in the skin after UVB irradiation. Journal of Photochemistry and Photobiology, B: Biology. 2012,109: 34-41.
  5. Alafiatayo, A.A.; Lai, K.-S.; Ahmad, S.; Mahmood, M.; Shaharuddin, N.A. RNA-Seq analysis revealed genes associated with UV-induced cell necrosis through MAPK/TNF-α pathways in human dermal fibroblast cells as an inducer of premature photoaging. Genomics. 2020, 112: 484–493.
  6. Berneburg, M.; Gattermann, N.; Stege, H.; Grewe, M.; Vogelsang, K.; Ruzicka, T.; Krutmann, J. Chronically ultraviolet-exposed human skin shows a higher mutation frequency of mitochondrial DNA as compared to unexposed skin and the hematopoietic system. Photochemistry and Photobiology. 1997, 66:271–275.
  7. Premi, S.; Brash, D.E. Chemical excitation of electrons: a dark path to melanoma. DNA Repair. 2016,169–177.
  8. Bedard. K.; Krause, K.H.The NOX family of ROS-generating NADPH oxidases: physiology and pathophysiology. Physiological Reviews. 2007, 87(1):245-313.
  9. Morry, J.; Ngamcherdtrakul, W.; Yantasee, W. Oxidative stress in cancer and fibrosis: Opportunity for therapeutic intervention with antioxidant compounds, enzymes, and nanoparticles. Redox Biology. 2017; 11(C):240-253.
  10. Chapela SP, Burgos I, Congost C, Canzonieri R, Muryan A, Alonso M, Stella CA. Parenteral Succinate Reduces Systemic ROS Production in Septic Rats, but It Does Not Reduce Creatinine Levels. Oxid Med Cell Longev. 2018 Nov 6;2018: 1928945. doi: 10.1155/2018/1928945.
  11. Carvour M, Song C, Kaul S, Anantharam V, Kanthasamy A, Kanthasamy A. Chronic low-dose oxidative stress induces caspase-3-dependent PKCdelta proteolytic activation and apoptosis in a cell culture model of dopaminergic neurodegeneration. Ann N Y Acad Sci. 2008 Oct; 1139:197-205. doi: 10.1196/annals.1432.020.

- Since ascorbic acid can only remove superoxide, it is doubtful whether it can be used as a control when measuring H2O2 this time.

Reply:

Thank you for your comments and great suggestions. H2O2-induced model establishment is commonly used in the study of oxidative damage. H2O2 is used as an inducer to establish H2O2 damage model, cells will produce a large number of ROS. In the study, we measured the cellular ROS contents to evaluate the oxidative damage level.

Ascorbic acid (VC) was chosen as a positive control in the manuscript. It has an excellent antioxidant effect and also has the function of scavenging ROS radicals [4,5]. Besides, some published articles in our laboratory also verified intracellular ROS scavenging activities [1-3].

1.Su, Y.; Zhang, Y.; Fu, H.; Yao, F.; Liu, P.; Mo, Q.; Wang, D.; Zhao, D.; Wang, C.; Li, M. Physicochemical and Anti-UVB-Induced Skin Inflammatory Properties of Lacticaseibacillus Paracasei Subsp. Paracasei SS-01 Strain Exopolysaccharide. FERMENTATION-BASEL 2022, 8, doi:10.3390/fermentation8050198.

2.Fu, H.; Zhang, Y.; An, Q.; Wang, D.; You, S.; Zhao, D.; Zhang, J.; Wang, C.; Li, M. Anti-Photoaging Effect of Rhodiola Rosea Fermented by Lactobacillus Plantarum on UVA-Damaged Fibroblasts. NUTRIENTS 2022, 14, doi:10.3390/nu14112324.

  1. 杜,Y.-T。长,Y。唐,W。刘 X.-F.;戴,F。Zhou, B. 药理学维生素 C 对 NF-κB 介导的炎症的促氧化抑制作用。自由基生物学和医学 2022, 180, 85–94, doi:10.1016 / j。freeradbiomed.2022.01.007。
  2. Kouakanou L, Xu Y, Peters C, He J, Wu Y, Yin Z, Kabelitz D. 维生素 C 促进人类 γδ T 细胞的增殖和效应功能。细胞摩尔免疫学。2020 年 5 月;17(5):462-4 doi: 10.1038/s41423-019-0247-8。Epub 2019 年 6 月 6 日。PMID:31171862;
  3. Zhang C, Zhou Y, Yang GY, Li S. 仿生肽保护细胞免受氧化应激。我是 J Transl Res。2017 年 12 月 15 日;9(12):5518-5527。

Reviewer 2 Report

Journal: Applied Sciences (ISSN 2076-3417)

Manuscript ID: applsci-1874334

Type: Article

Title: Ectoin Exerts Dermatoprotective Properties against UVA/H2O2-Induced Damage caused to Human Skin Fibroblast Cells via the Activation of the PI3K/AKT Pathway.

Authors: Wenjing Cheng, Quan An, Jiachan Zhang *, Xiuqin Shi, Changtao Wang, Meng Li, Dan Zhao.

a)           Line 45: [5-7].

b)          Line 50: [11-14].

c)           Line 54: [15-17].

d)          Figures 2, 3, 4 and 5 write legend on X-axis.

e)           For references, choose recent refs. refer to these refs.

1-          Mustafa Nuhad Al-Darraji, Saade abdalkareem Jasim, Omer Dhia Aldeen Salah Aldeen, Abdolmajid Ghasemian, Mohammed Rasheed “The effect of LL37 antimicrobial peptide on FOXE1 and lncRNA PTCSC 2 genes expression in colorectal cancer (CRC) and normal cells”, Asian Pacific Journal of Cancer Prevention, 2022 to appear.

2-          Mustafa Nuhad Al-Darraji, Liqaa Hasson Saqban, Thulfiqar Fawwaz Mutar, Mohammed Rasheed, Aasim Jasim Hussein, “Association of Candidate Gene Polymorphisms in Iraqi Patients with Chronic Kidney Disease”, J Adv Biotechnol Exp Ther. 2022; 5(3): 687-701.

Best Regards

Author Response

Detailed response to reviewers’ comments

Manuscript ID: applsci-1874334

The revised title: Protective effect of Ectoin on UVA/H2O2-induced Oxidative Damage in Human Skin Fibroblast Cells

Authors: Wenjing Cheng, Quan An, Jiachan Zhang *, Xiuqin Shi, Changtao Wang *, Meng Li, Dan Zhao

Dear reviewers,

First of all, we would like to thank the reviewers for their comments on revising the manuscript, and we are also very grateful to the reviewer for their work on our manuscript. Detailed point-by-point responses to the reviewers’ comments are provided in the following pages. Note that the reviewers’ comments are presented in Italics, and our responses are in Roman and blue font. In addition, we addressed all these major points and other issues carefully and revised the manuscript accordingly, and We highlight the revised parts in the article in red. Please let me know if you have any further questions.

Sincerely,

Jiachan Zhang

Beijing Key Lab of Plant Resource Research and Development, Beijing Technology and Business University, Fucheng Road, Beijing 100048, China

Tel.: +86-13426258535

To the referee’s comments, we make the following responses and changes in the manuscript:

- Line 45: [5-7]; Line 50: [11-14]; Line 54: [15-17].

Reply: Thank you for your comments and great suggestions. We have checked and revised this part in the manuscript. Thanks again for your advice, which is very useful for our manuscript.

-Figures 2, 3, 4 and 5 write legend on X-axis.

Reply: Thank you for your comments. We have checked and revised this part in the Figures. Thanks again for your advice, which is very useful for our manuscript.

 -For references, choose recent refs. refer to these refs. a)        

Reply:

Thank you for your comments. We have updated the references in the manuscript. We are very sorry that we did not find the two articles recommended by the reviewer due to the lack of online resources. We also sought help from other institutions, but we also failed to find these two articles. If possible, we hope that the reviewer can send us these two articles so that we can learn the relevant content. Thank you again for your suggestion, which is very important for us to supplement and revise the manuscript.

Reviewer 3 Report

The authors of the present study built Ultra Violet A- and Hydrogen peroxide-induced oxidation models of human skin fibroblasts for investigation of the protective effects of Ectoin. The article can be published after following corrections:

General remarks: The language of the manuscript must be revised. It contains lots of writing mistakes (particularly names of chemical compounds).

Title: The title of the study must be revised. The authors have not studied the effects of UVA, H2O2 and Ectoin on PI3K/AKT pathway. The title must not contain PI3K/AKT pathway.

Abstract: These sentences need revision as the authors did not study PI3K/AKT pathway. “However, only few reports have been published on the protective effect exerted by Ectoin against oxidative damage, especially the protective effect of Ectoin exerted on the PI3K/AKT pathway”.

“Further studies on the mechanisms involved, for exam-26 ple, the expression levels of genes or proteins associated with the PI3K/AKT signaling pathway and 27 levels of antioxidant enzymes in cells were determined”.

“In conclusion, 31 our results indicate that Ectoin exerts dermatoprotective properties by accelerating the PI3K/AKT 32 signaling pathway and upregulating antioxidative enzymes levels.”

Introduction:

a)     The authors should include following sentences after line 50.

“Protecting the tissues from unwanted side effects is one strategy to lessen the unfavorable consequences of UVA and H2O2 in medical practice. As a result, the therapeutic benefits of natural compounds with antioxidant capabilities that may lessen the severity of UVA and H2O2 -induced toxicities could be advantageous. Antioxidants are widely utilized as nutrients, and their effectiveness in reducing tissue and organ toxicity from various medicines was studied. Numerous studies have looked at the effectiveness of antioxidants as nutrients in reducing tissue and organ toxicity caused by a variety of conditions. Reactive oxygen species damage can be avoided and repaired by endogenous and exogenous antioxidants (ROS). They are referred to as "free radical scavengers" because they can strengthen the immune system and reduce the danger of sickness and toxins. Superoxide and other peroxides are chelated by enzyme-based antioxidants such as catalase (CAT), glutathione peroxidase (GPx), and superoxide dismutase (SOD). They serve as defense mechanisms against endogenous antioxidants, clearing ROS activity and buildup in cells and preserving redox balance (1-4).

 Zengin, G., Mahomoodally, M. F., Aktumsek, A., Jekő, J., Cziáky, Z., Rodrigues, M. J., ... & Picot-Allain, C. (2021). Chemical profiling and biological evaluation of Nepeta baytopii extracts and essential oil: An endemic plant from Turkey. Plants, 10(6), 1176.

b)    Revise the sentences in line between 83-86 as suggested above (PI3K/AKT pathway)

Material and Method and Results: No correction is needed

Discussion: Some mechanistic explanations related effects of UVA, H2O2 and ectoin is needed in discussion. The authors should revise their discussion based on similar reasons mentioned above.

Conclusion: Must be improved.

Author Response

Detailed response to reviewers’ comments

Manuscript ID: applsci-1874334

The revised title: Protective effect of Ectoin on UVA/H2O2-induced Oxidative Damage in Human Skin Fibroblast Cells

Authors: Wenjing Cheng, Quan An, Jiachan Zhang *, Xiuqin Shi, Changtao Wang *, Meng Li, Dan Zhao

Dear reviewers,

First of all, we would like to thank the reviewers for their comments on revising the manuscript, and we are also very grateful to the reviewer for their work on our manuscript. Detailed point-by-point responses to the reviewers’ comments are provided in the following pages. Note that the reviewers’ comments are presented in Italics, and our responses are in Roman and blue font. In addition, we addressed all these major points and other issues carefully and revised the manuscript accordingly, and We highlight the revised parts in the article in red. Please let me know if you have any further questions.

Sincerely,

Jiachan Zhang

Beijing Key Lab of Plant Resource Research and Development, Beijing Technology and Business University, Fucheng Road, Beijing 100048, China

Tel.: +86-13426258535

To the referee’s comments, we make the following responses and changes in the manuscript:

-General remarks: The language of the manuscript must be revised. It contains lots of writing mistakes (particularly names of chemical compounds).

Reply:

Thank you for your comments and great suggestions. We have checked and revised the language of the article, especially the compound name. Thank you again for pointing out our problem, which is very helpful for us to revise the manuscript.

-Title: The title of the study must be revised. The authors have not studied the effects of UVA, H2O2 and Ectoin on PI3K/AKT pathway. The title must not contain PI3K/AKT pathway.

Reply:

Thank you for your comments and great suggestions. We have revised the title in the manuscript, and the new title is 'Protective Effect of Ectoin on UVA/H2O2-induced Oxidative Damage in Human Skin Fibroblast Cells'.

Thank you again for your suggestions, which are very important for us to improve this manuscript.

-Abstract: These sentences need revision as the authors did not study PI3K/AKT pathway. “However, only few reports have been published on the protective effect exerted by Ectoin against oxidative damage, especially the protective effect of Ectoin exerted on the PI3K/AKT pathway”.

Further studies on the mechanisms involved, for exam-26 ple, the expression levels of genes or proteins associated with the PI3K/AKT signaling pathway and 27 levels of antioxidant enzymes in cells were determined”.

In conclusion, 31 our results indicate that Ectoin exerts dermatoprotective properties by accelerating the PI3K/AKT 32 signaling pathway and upregulating antioxidative enzymes levels.”

  Reply:

Thank you for your comments and great suggestions. We have checked and revised the Abstract in the manuscript. Thank you again for your suggestions, which are very important for us to improve this manuscript. We have made some changes as shown below, which were also marked in red in the revised manuscript.

 However, there are few reports on the protective effect of Ectoin on oxidative damage, especially on the regulation of PI3K/AKT pathway related genes at mRNA level.

 Further studies on the mechanisms involved, for example, the expression levels of genes associated with the PI3K/AKT signaling pathway and levels of antioxidant enzymes in cells were determined.

 In conclusion, our results indicate that Ectoin exerts dermatoprotective properties by upregulated genes COL1A1, COL1A2, FN1, IGF2, NR4A1, and PIK3R1 and upregulating antioxidative enzymes levels.

-Introduction:

  1. a) The authors should include following sentences after line 50.

Protecting the tissues from unwanted side effects is one strategy to lessen the unfavorable consequences of UVA and H2O2 in medical practice. As a result, the therapeutic benefits of natural compounds with antioxidant capabilities that may lessen the severity of UVA and H2O2 -induced toxicities could be advantageous. Antioxidants are widely utilized as nutrients, and their effectiveness in reducing tissue and organ toxicity from various medicines was studied. Numerous studies have looked at the effectiveness of antioxidants as nutrients in reducing tissue and organ toxicity caused by a variety of conditions. Reactive oxygen species damage can be avoided and repaired by endogenous and exogenous antioxidants (ROS). They are referred to as "free radical scavengers" because they can strengthen the immune system and reduce the danger of sickness and toxins. Superoxide and other peroxides are chelated by enzyme-based antioxidants such as catalase (CAT), glutathione peroxidase (GPx), and superoxide dismutase (SOD). They serve as defense mechanisms against endogenous antioxidants, clearing ROS activity and buildup in cells and preserving redox balance (1-4).

 Zengin, G., Mahomoodally, M. F., Aktumsek, A., Jekő, J., Cziáky, Z., Rodrigues, M. J., ... & Picot-Allain, C. (2021). Chemical profiling and biological evaluation of Nepeta baytopii extracts and essential oil: An endemic plant from Turkey. Plants, 10(6), 1176.

Reply:

Thank you for your comments and great suggestions. We have added this in the Introduction. Besides, thanks again for the literature recommended by the reviewer, which made us more familiar with this field. Meanwhile, we also read other relevant literatures to support this statement.

  1. b) Revise the sentences in line between 83-86 as suggested above (PI3K/AKT pathway)

Reply:

Thank you for your comments and great suggestions. We have modified these sentences in the Introduction and marked in red, which was also shown as follows:

In this study, we established UVA-induced and H2O2-induced oxidative models of human skin fibroblast cells separately, discuss the protective effects exerted by Ectoin in the two models, and further studied the molecular mechanisms.

-Discussion: Some mechanistic explanations related effects of UVA, H2O2 and ectoin is needed in discussion. The authors should revise their discussion based on similar reasons mentioned above.

Reply:

Thank you for your comments and great suggestions. We have checked and revised this part of the Discussion, which we marked in red in the revised manuscript. Thank you again for your suggestions, which are very important for us to improve this manuscript.

Round 2

Reviewer 3 Report

The article can be published in its current form